# Ocean liming effects on dissolved organic matter dynamics

Chiara Santinelli[1], Silvia Valsecchi[1,2,3], Simona Retelletti Brogi[1,4], Giancarlo Bachi[1], Giovanni Checcucci[1], Mirco Guerrazzi[1], Elisa Camatti[5], Stefano Caserini[3,6], Arianna Azzellino[2,3], Daniela Basso [3,7]

Consiglio Nazionale delle Ricerche (CNR), Istituto di Biofisica. Via Moruzzi 1, 56124 Pisa (PI), Italia.
Politecnico di Milano, Dipartimento di Ingegneria Civile ed Ambientale. Piazza Leonardo da Vinci 32, 20133 Milano (MI), Italia.
Consorzio Nazionale Interuniversitario per le Scienze del Mare (CoNISMa). Piazzale Flaminio 9, 00196 Roma (RM), Italia.
Istituto di Oceanografia e Geofisica Sperimentale (OGS), Sezione di Oceanografia. Via Piccard 54, 34151 Trieste (TS), Italia.
Consiglio Nazionale delle Ricerche (CNR), Istituto di Scienze Marine. Arsenale Tesa 104, Castello 2737/F - 30122 Venezia (VE), Italia.
Università di Parma, Dipartimento di Ingegneria e Architettura. Parco Area delle Scienze 181/A, 43124 Parma (PR), Italia.
Università degli Studi di Milano-Bicocca, Dipartimento di Scienze dell'ambiente e della terra. Piazza della Scienza 4, 20126 Milano (MI), Italia.

*Correspondence to*: Chiara Santinelli (chiara.santinelli@ibf.cnr.it)

**Abstract**. Ocean liming has gained attention as a potential solution to mitigate climate change by actively removing
carbon dioxide ($CO_2$) from the atmosphere. The addition of hydrated lime into oceanic surface water leads to an increase
in alkalinity, which in turn promotes the uptake and sequestration of atmospheric $CO_2$.
Despite the potential of this technique, its effects on the marine ecosystem are still far to be understood, and there is
currently no information on the potential impacts on the concentration and quality of Dissolved Organic Matter (DOM),
that is one of the largest, the most complex and yet the least understood mixture of organic molecules on Earth.
The aim of this study is to provide the first experimental evidence about the potential effects of hydrated lime addition on
DOM dynamics in the oceans, by assessing changes in its concentration and optical properties (absorption and
fluorescence).
To investigate the effects of liming on DOM pools with different concentrations and quality, seawater was collected from
two contrasting environments: the oligotrophic Mediterranean Sea, known for its Dissolved Organic Carbon (DOC)
concentration comparable to that observed in the oceans, and the eutrophic Baltic Sea, characterized by high DOM
concentration mostly of terrestrial origin. hydrated lime was added in both waters, to reach a pH of 9 and 10.
Our findings reveal that the addition of hydrated lime has a noticeable effect on DOM dynamics in both the Mediterranean
Sea and Baltic Sea, determining a reduction in DOC concentration and a change in the optical properties (absorption and
fluorescence) of DOM. These effects, detectable at pH 9, become significant at pH 10 and are more pronounced in the
Mediterranean Sea than in the Baltic Sea. These potential short-term effects should be considered within the context of
the physico-chemical properties of seawater and the seasonal variability.

## 1 Introduction

Oceans are a natural sink for atmospheric $CO_2$ having the potential to mitigate its increase and therefore the effects of climate change (Gattuso et al., 2013; Heinze et al., 2015). The massive amount of atmospheric $CO_2$ absorbed by the oceans in the last decades (~ 30-40% of anthropogenic emissions), is generating dramatic global-scale changes in seawater chemistry, such as a decrease in pH, in carbonate concentration and in the ocean buffering capacity (Chikamoto et al., 2023). Even if the ongoing efforts toward a global reduction of anthropogenic $CO_2$ emissions should be rapidly intensified, the available projections highlight the need for additional strategies, such as the development of efficient ocean-based Negative Emission Technologies (NETs) (Calvin et al., 2023; Royal Society and Royal Academy of Engineering, 2018). Some NETs are not only capable of removing atmospheric $CO_2$ and store it as bicarbonate ions into the oceans, but also of increasing the water pH, restoring ocean buffering capacity to the pre-industrial era (Butenschön et al., 2021; Gore et al., 2019). One of these NETs is Ocean Alkalinity Enhancement (OAE) (also called Artificial Ocean Alkalinization, AOA), which relies on the dissolution of alkaline minerals such as hydrated lime (calcium hydroxide, $Ca(OH)_2$) into the oceans) (Kheshgi, 1995). Although the exact amount of hydrated lime to be released, as well as its sparging methods, is still under debate one of the proposals is to discharge highly concentrated slurry (*lime milk*) from large cargo ships, tankers and/or dedicated vessels. Caserini et al., (2021) simulated the pH dynamics within the wake of a sparging ship releasing $Ca(OH)_2$ with an initial particle radius of 45 μm at a rate of 10 kg s$^{-1}$. The results of their modeling study suggest that in these conditions a temporary, sharp increase in pH of about 1 unit can be observed at the discharge site, and that the effects decrease moving far from the discharge site, becoming lower than 0.2 pH units at a distance of 1400 – 1600 m (0.8-0.9 nautical miles).

The discharge of alkaline minerals may trigger the inorganic precipitation of calcium carbonate ($CaCO_3$), reducing the efficiency of the $CO_2$ sink and negatively affecting seawater transparency and photosynthetic rates (González and Ilyina, 2016), with possible consequences for the biogeochemical cycles and the functioning of the marine ecosystem (Camatti et al., 2024). The side effects of OAE techniques on the marine environment need to be thoroughly investigated before making any decision on their use. To the best of our knowledge, there is no information on the effects that ocean liming may have on Dissolved Organic Matter (DOM) and its chromophoric fraction (CDOM, i.e. the light-absorbing fraction and FDOM, i.e. its fluorescent fraction). Holding an amount of carbon of 660 billion metric tons and being the most concentrated dissolved component in the oceans (Hansell et al., 2009), every action that could modify seawater chemistry is expected to have an impact on this key component of the carbon cycle. DOM represents the main source of energy for heterotrophic prokaryotes, a change in its concentration and/or quality could therefore have a cascading effect on the functioning of marine ecosystem.

The aim of this study is to provide the first experimental evidence about the potential effects of hydrated lime addition on DOM dynamics in the oceans, by assessing changes in its concentration and optical properties (absorption and fluorescence). In order to investigate the impact on DOM pool with different origin and optical properties, seawater was collected from two highly diverse environments; (1) the oligotrophic Mediterranean Sea, characterized by Dissolved Organic Carbon (DOC) concentration comparable to those observed in the open oceans, and (2) the eutrophic Baltic Sea, characterized by high DOC concentration, mostly of terrestrial origin.

## 2 Materials and methods

In order to investigate the effects of ocean liming on DOM dynamics, an ultra-pure $Ca(OH)_2$ powder was added to natural seawater and changes in DOC concentration, absorption and fluorescence of CDOM were followed for 24 hours at the laboratories of the Biophysics Institute, CNR (Pisa, Italy). Based on the results by Caserini et al. (2021), which suggested a sharp increase of 1 unit of pH at the discharge site of a sparging ship, the experiment was carried out at pH 9. Although unlikely under actual conditions of dilution in the open sea, an additional experiment was carried out at pH 10 because this situation may occur in coastal waters (e.g. coastal lagoons, high primary productivity enhanced by eutrophication) (Hinga, 2002). $Ca(OH)_2$ was provided by UNICALCE (Sedrina (BG), Italy) and supplied as powder (Tab. S1). Seawater was collected at Marina di Pisa, Tyrrhenian Sea, Italy (Mediterranean Sea) and in the coastal area surrounding Riga, Latvia (Baltic Sea) (Tab. 1).

|  | Sampling Date | Salinity | pH | DOC ($\mu M$) | $a_{254}$ ($m^{-1}$) | $S_{275-295}$ ($nm^{-1}$) |
|---|---|---|---|---|---|---|
| **Mediterranean Sea** | Mar-22 | 38 | 8.2 | $66 \pm 0.5$ | 1.9 | 0.024 |
| **Baltic Sea** | Apr-22 | 6 | 8.1 | $364 \pm 3$ | 24.5 | 0.021 |

**Table 1: Chemical and physical properties of the Mediterranean Sea and Baltic Sea water used for the experiment.**

## 2.1 Experimental setup

In order to investigate the impact of slaked lime on chemical-physical processes affecting DOM dynamics, seawater was sterilized by filtration through a 0.2 µm pore size filter (Polycap AS36 filter capsule, Whatman, UK) using a peristaltic pump (Masterflex™ L/S™, Germany). Salinity was measured by using a HI 9033 portable probe (Hanna Instruments, USA). The experiments were carried out in 2 L acid-washed polycarbonate ®Nalgene bottles as follows:

1. **Mediterranean Sea**
   a. Treatment: filtered surface seawater enriched with $Ca(OH)_2$ powder to reach:
      - pH 9, $[Ca(OH)_2]$ 0.04 g/L
      - pH 10, $[Ca(OH)_2]$ 0.25 g/L
   b. Control: filtered surface seawater (pH = 8.2)
2. **Baltic Sea**
   a. Treatment: filtered surface seawater enriched with $Ca(OH)_2$ powder to reach
      - pH of 9, $[Ca(OH)_2]$ 0.01 g/L
      - pH 10, $[Ca(OH)_2]$ 0.06 g/L
   b. Control: filtered surface seawater (pH = 8.1)

All the experiments were carried out in triplicates and the bottles were stored in the dark and at room temperature ($22 \pm 1$ °C). Immediately after the addition of the $Ca(OH)_2$ powder, the bottles were gently mixed. Before and after powder addition and before each sampling time, pH was measured using an edge HI2002-02 pH-meter (Hanna Instruments, USA).

In the treatment at pH 9, the pH slightly decreased by 0.06 (Baltic Sea) and 0.29 (Mediterranean Sea) between 3 and 22
h after the addition (Tab. S2). In the treatment at pH 10, 3 hours after the addition the pH decreased by 0.3 in the
Mediterranean Sea and after 22 hours it decreased by 0.45 in the Mediterranean Sea and 0.26 in the Baltic Sea (Tab. S2).
Subsamples for DOC (40 mL) and CDOM/FDOM (60 mL) analyses were collected before Ca(OH)$_2$ addition and 5', 30',
3 h and 22 h after Ca(OH)$_2$ addition.
The bottles were gently mixed before subsampling at 5', 30', and 3 h. After 22 hours, carbonate sedimentation was clearly
visible at the bottom of the bottles, samples of the supernatant were therefore collected before mixing for both DOC and
CDOM/FDOM analyses, and an additional sample was collected after gently mixing only for DOC analyses since
CDOM/FDOM would have been strongly affected by the scattering due to the suspended particles.
Samples for CDOM/FDOM analyses were brought to pH 7.5 ± 1.0 with high purity 2 M HCl, to avoid the effect of pH
on DOM absorption and fluorescence and filtered through a PES 0.2 µm pore size syringe filter (Minisart 16534K,
Sartorius, Germany), to avoid the scattering due to the presence of carbonate particles in solution.

## 2.2 DOC

Samples for DOC analyses were acidified at pH 2 with high purity 2 M HCl. DOC measurements were carried out with
a TOC-L analyzer (Shimadzu, Japan), by high temperature catalytic oxidation following Santinelli et al. (2015). Samples
were sparged for 3 min with CO$_2$-free ultrahigh purity nitrogen to remove inorganic carbon. 150 µL of the sample were
injected into the furnace after a three-fold rinsing with the sample to be analyzed. From 3 to 5 replicate injections were
performed until the analytical error was lower than 1%. A four-point calibration curve was measured using a standard
solution of potassium hydrogen phthalate in the same concentration range as the samples. The system blank was measured
every day at the beginning and the end of the analyses using low-carbon water (2-3 µM C). The instrument performance
was verified daily using the DOC Consensus Reference Material (CRM) (Hansell, 2005) (CRM Batch #20/08-20, nominal
concentration of 42 ± 1 µM; measured concentration 40 ± 2 µM, 76 CRM samples analyzed).

## 2.3 CDOM optical properties

### 2.3.1 Absorption

Absorption spectra were measured between 230 and 700 nm with a UV–Vis spectrophotometer (Mod-7850, Jasco, USA),
using a 10 cm quartz cuvette. The absorption spectrum of Milli-Q water was subtracted from each sample spectrum. The
absorption coefficient at 254 nm (a$_{254}$) and the spectral slope between 275 and 295 nm (S$_{275-295}$) were calculated from the
absorption spectra using the ASFit tool (Omanović et al., 2019). a$_{254}$ is used to have semi-quantitative information on
CDOM, since primary CDOM absorption is caused by conjugated systems having the absorption peak near 254 nm (Del
Vecchio and Blough, 2004; Weishaar et al., 2003). S$_{275-295}$ can be related to changes in the average aromaticity and
molecular weight of the molecules in the CDOM pool (Helms et al., 2008).The absorption coefficient at 280 nm and 325
nm (a$_{280}$, a$_{325}$) and the spectral slope ratio (Sr, ratio between S$_{275-295}$ and S$_{350-400}$) are also reported for comparison (Tab,
S2), being among the most used CDOM indices in the literature.

### 2.3.2 Fluorescence

Fluorescence excitation-emission matrices (EEMs) were recorded with an Aqualog fluorometer (Horiba-Jobin Yvon, UK), using a 1 cm quartz cuvette. Excitation ranged between 250 and 450 nm at 5 nm increments; emission was recorded between 212 and 619 nm at 3 nm increments. The EEMs were processed using the TreatEEM software (Omanović et al., 2023). EEMs were corrected for instrumental bias in excitation and emission, and Rayleigh and Raman scatter peaks were removed using the monotone cubic interpolation (shape-preserving). EEMs were normalized to the water Raman signal, dividing the fluorescence by the integrated Raman band of Milli-Q water ($\lambda_{ex}$ = 350 nm, $\lambda_{em}$= 371-428 nm; Lawaetz and Stedmon, 2009) measured on the same day of the analyses. The fluorescence intensity is therefore reported as equivalent to water Raman Units (R.U.).

Parallel factor analysis (PARAFAC) was separately applied to the Mediterranean Sea and Baltic Sea samples (number of EEMs: 45 for each experiment), using the decomposition routines of the EEMs toolbox for MATLAB software (drEEM) (Murphy et al., 2013). The PARAFAC validated a 3-component model for both the Mediterranean Sea and the Baltic Sea (Fig. S1 and S2). OpenFluor, an online database of environmental fluorescence spectra, was used as a validation tool to characterize the three components (Tab. S3 and S4). OpenFluor compares excitation and emission spectra of the validated components with all the components present in the database and allows comparing the spectra using the Tucker Congruence Coefficient (TCC; Murphy et al. (2014)).

### 2.4 Statistical analyses

For all parameters, differences were tested using the Kruskal–Wallis nonparametric test and were considered significant at the threshold of $p < 0.05$. All statistical analyses were performed using OriginPro version 9 (OriginLab, USA).

### 3. Results

### 3.1 Mediterranean Sea

### 3.1.1 DOC

In the Mediterranean Sea, three hours after Ca(OH)$_2$ addition, a 4 µM (6%) DOC decrease was observed in both treatments (Fig. 1, Tab. S2). A further decrease was observed in the supernatant of the unmixed sample 22 h after the addition, with DOC reaching $59.5 \pm 0.1$ µM (9% decrease) at pH 9 and $56.3 \pm 1.7$ µM (18% decrease) at pH 10 (Tab. S2). It is noteworthy that such a decline was only observed in the unmixed samples, whereas no significant change was observed in the mixed samples 22 h after the addition (Fig.1, Tab. S2)

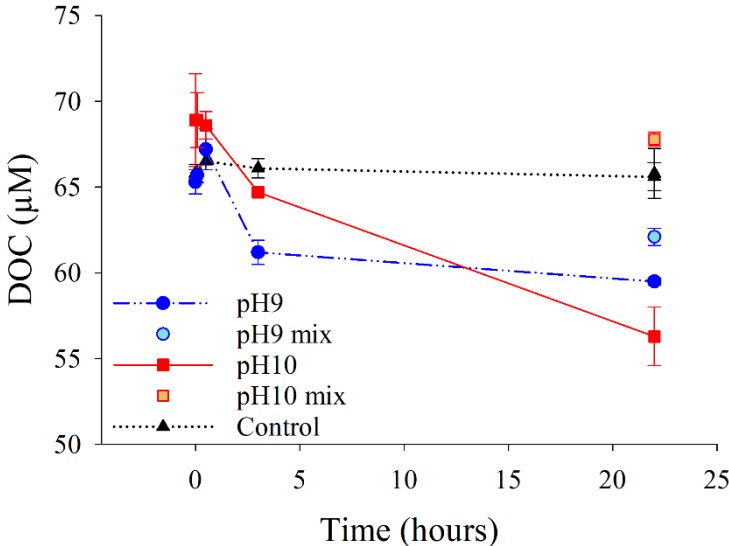

168

**Figure 1: Trend of DOC concentration in the Mediterranean Sea treatments at pH 9 and 10, and in the control. Error bars refer to the standard deviation among the 3 replicates. Please, note that for some samples the error bars are smaller than the symbols and therefore not visible.**

**3.1.2 CDOM Absorption**

A slight decrease in $a_{254}$ was observed 3 hours after Ca(OH)$_2$ addition at both pH (9 and 10). Interestingly, 22 h after the addition, in the supernatant of unmixed bottles, a marked decrease of 0.2 m$^{-1}$ (10%) and 0.4 m$^{-1}$ (19%) was observed (Fig. 2a, Tab. S2) together with an increase in $S_{275-295}$ from 0.0247 nm$^{-1}$ to 0.0256 nm$^{-1}$ (4%) and from 0.0239 to 0.0264 nm$^{-1}$ (10%) at pH 9 and 10, respectively (Fig. 2b, Tab. S2). Mixed samples were not collected for CDOM analyses, since scattering due to the particles would have affected the results.

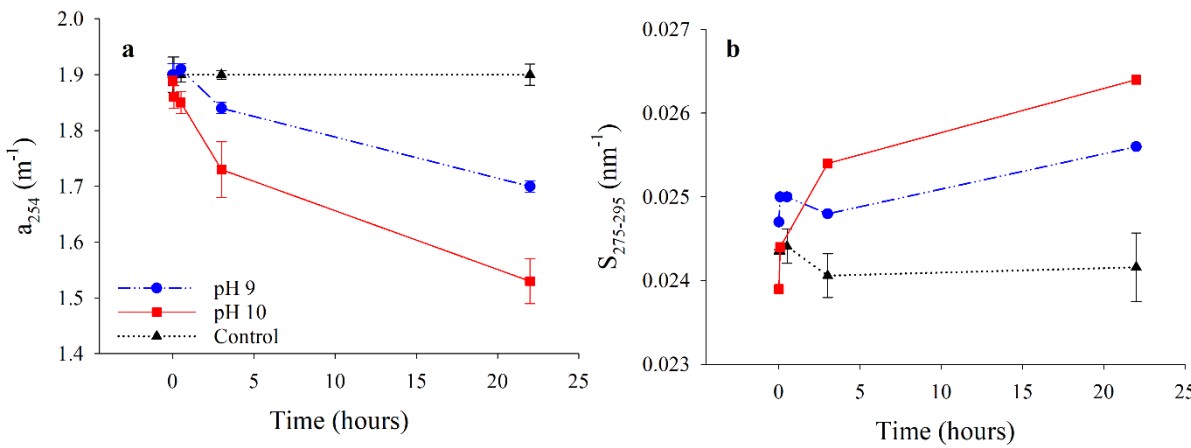

180

**Figure 2: Trend of $a_{254}$ (a) and $S_{275-295}$ (b) in the Mediterranean Sea treatments at pH 9 and 10, and in the control. Error bars refer to the standard deviation among the 3 replicates. Please, note that for some samples the error bars are smaller than the symbols and therefore not visible.**

### 3.1.3 FDOM fluorescence

The PARAFAC validated a 3-component model for the Mediterranean Sea EEMs (Fig. S1, Tab. S3). Component 1 ($\lambda_{Ex}/\lambda_{Em}$: 315/409 nm, Fig. S1a) shows spectroscopic characteristics similar to Coble's peak M ($\lambda_{Ex}/\lambda_{Em}$: 312 /[380]420; Coble (1996)). The comparison with similar components in the OpenFluor database (matches with a TCC > 0.95) allowed to characterize it as marine humic-like ($C1_{Mh-Med}$). Component 2 ($\lambda_{Ex}/\lambda_{Em}$: 275/331 nm, Fig. S1b) shows spectroscopic characteristics similar to Coble's peak T ($\lambda_{Ex}/\lambda_{Em}$: 275/340 nm; Coble (1996)). The comparison with similar components in the OpenFluor database (matches with a TCC > 0.95) allowed to characterize it as tryptophan-like ($C2_{Trp-Med}$). Component 3 ($\lambda_{Ex}/\lambda_{Em}$: 260/[380]456 nm, Fig. S1c) shows spectroscopic characteristics similar to Coble's peaks C and A ($\lambda_{Ex}/\lambda_{Em}$: 350/451 and 245/451 nm, respectively; Coble (1996)). The comparison with similar components in the OpenFluor database (matches with a TCC > 0.95) allowed to characterize it as terrestrial humic-like ($C3_{Th-Med}$).

$C1_{Mh-Med}$ did not show significant changes over the incubation time at pH 9 and in the control (Fig. 3a, Tab. S5). At pH 10, a decrease of 0.004 R.U. (20%) was observed 22 h after the addition. $C2_{Trp-Med}$ did not show significant changes during the incubation, neither in the treatments nor in the control (Fig. 3b, Tab. S5). $C3_{Th-Med}$ showed variations only at pH 10 (Fig. 3c, Tab. S5), with a slight decrease 3 hours after the addition, and a significant decrease of 0.006 R.U. (26%) at the end of the incubation (22 h).

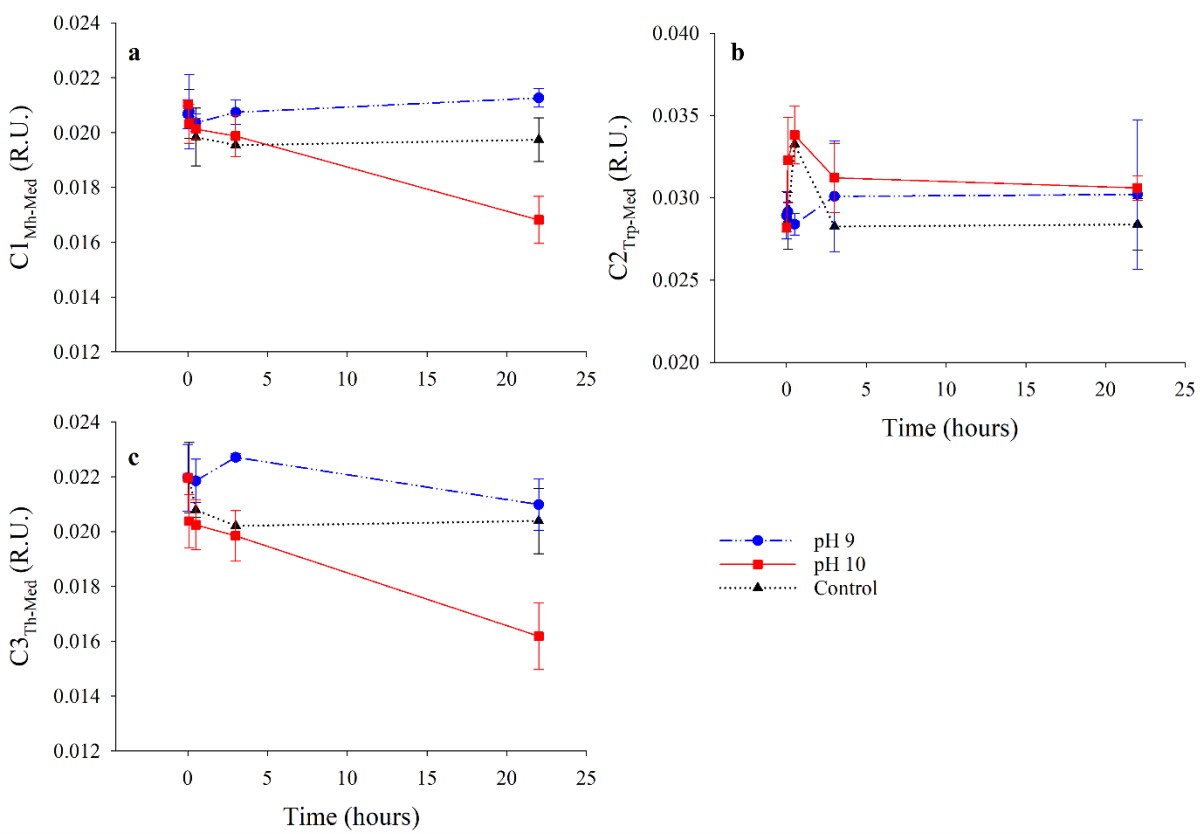

**Figure 3: Trend of the fluorescent intensity of C1$_{Mh-Med}$ (a), C2$_{Trp-Med}$ (b) and C3$_{Th-Med}$ (c) in the Mediterranean Sea treatments at pH 9 and 10, and in the control. Error bars refer to the standard deviation among the 3 replicates.**

205

## 3.2 Baltic Sea

### 3.2.1 DOC

In the Baltic Sea no significant change was observed 3 hours after $Ca(OH)_2$ addition in both treatments (pH 9 and 10) (Fig. 4, Tab. S2). At the end of the experiment (22 h), DOC decreased by 23 µM (6 %) at pH 10, whereas no significant change was observed at pH 9. It is noteworthy that DOC showed significant differences between the mixed and unmixed samples at pH 10, whereas in the mixed samples DOC was similar to the control (Fig. 4, Tab. S2).

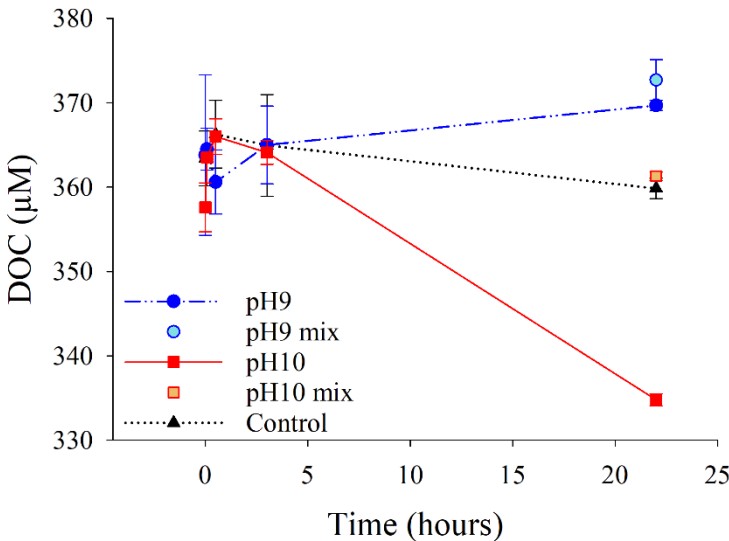

212

**Figure 4: Trend of DOC concentration in the Baltic Sea treatments at pH 9 and 10 and in the control. Error bars refer to the standard deviation among the 3 replicates. Please, note that for some samples the error bars are smaller than the symbols and therefore not visible.**

216

### 3.2.2 CDOM absorption

Twenty-two hours after the addition of $Ca(OH)_2$, $a_{254}$ decreased by 0.03 $m^{-1}$ (0.1%), and 3.6 $m^{-1}$ (15%) at pH 9 and pH 10, respectively (Fig. 5a). $S_{275-295}$ increased from 0.0215 $nm^{-1}$ to 0.0227 $nm^{-1}$ (6%) at pH 10, whereas no significant change was observed at pH 9 (Fig. 5b). The change in CDOM is therefore visible only at pH 10 (Fig. 5)

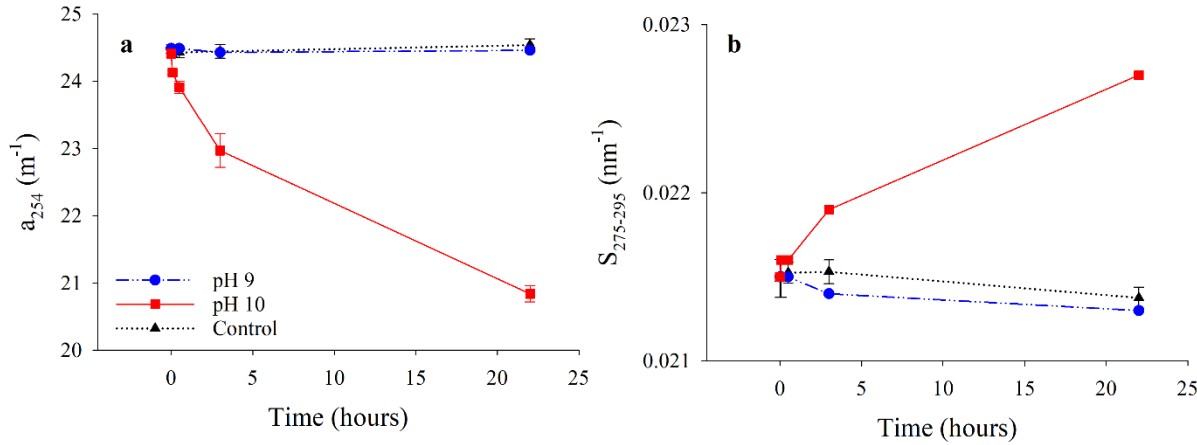

221

**Figure 5: Trend of $a_{254}$ (a) and $S_{275–295}$ (b) in the Baltic Sea treatments at pH 9 and 10, and in the control. Error bars refer to the standard deviation among the 3 replicates. Please, note that for some samples the error bars are smaller than the symbols and therefore not visible.**

225

226

### 3.2.3 FDOM fluorescence

The PARAFAC validated a 3-component model for the Baltic Sea EEMs experiment (Fig. S2, Tab. S3). Component 1 ($\lambda_{Ex}/\lambda_{Em}$: 290/400 nm, Fig. S2) shows spectroscopic characteristics similar to Coble's peak M ($\lambda_{Ex}/\lambda_{Em}$: 312 /[380]420; Coble (1996)). The comparison with similar components in the OpenFluor database (matches with a TCC > 0.95) allowed to characterize it as marine humic-like (C1$_{Mh-Bal}$). Component 2 ($\lambda_{Ex}/\lambda_{Em}$: 330/452 nm, Fig. S2) shows spectroscopic characteristics similar to Coble's peak C ($\lambda_{Ex}/\lambda_{Em}$: 350/451; Coble (1996)). The comparison with similar components in the OpenFluor database (matches with a TCC > 0.95) allowed to characterize it as Fulvic-like (C2$_{Flv-Bal}$). Component 3 ($\lambda_{Ex}/\lambda_{Em}$: 280/485 nm, Fig. S2) shows spectroscopic characteristics similar to Coble's peak A ($\lambda_{Ex}/\lambda_{Em}$: 260/[380]460 nm; Coble (1996)). The comparison with similar components in the OpenFluor database (matches with a TCC > 0.95) allowed to characterize it as Terrestrial humic-like (C3$_{Th-Bal}$).

C1$_{Mh-Bal}$ did not show significant changes during the incubation neither in the treatments nor in the control (Fig. 6a, Tab. S5). C2$_{Flv-Bal}$ did not show significant changes during the incubation at pH 9 and in the control, whereas a decrease of 0.03 R.U. (10%) was observed at pH 10 after 22 hours (Fig. 6b, Tab. S5). C3$_{Th-Bal}$ did not show significant changes during the incubation at pH 9 and in the control, whereas a decrease of 0.05 R.U. (13%) was observed at pH 10 after 22 hours (Fig. 6c, Tab. S5).

242

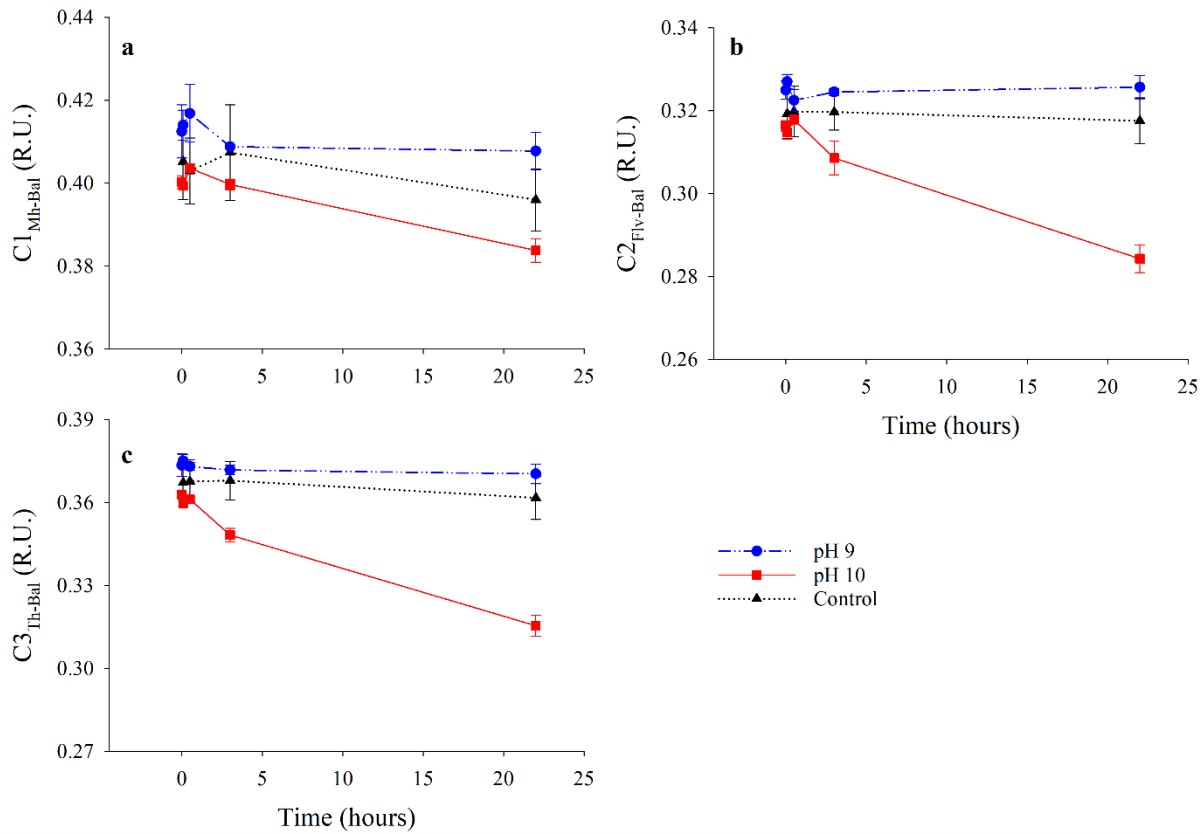

**Figure 6: Trend of the fluorescent intensity of $C1_{Mh-Bal}$ (a), $C2_{Flv-Bal}$ (b) and $C3_{Th-Bal}$ (c) in the Baltic Sea treatments at pH 9 and 10, and in the control. Error bars refer to the standard deviation among the 3 replicates. Please, note that for some samples the error bars are smaller than the symbols and therefore not visible.**

## 4 Discussion

Even if OAE using alkaline minerals is considered a promising tool to mitigate climate change through the sequestration and storage of atmospheric $CO_2$ into the ocean (DOSI, 2022), its impact on the marine ecosystem is still far to be understood. To the best of our knowledge, this is the first study investigating the potential effects of OAE by hydrated lime addition on DOM dynamics, with particular regard to DOC concentration and CDOM optical properties. Given the crucial role that DOM plays in the marine ecosystem, any impact on its dynamics is expected to affect the water quality and ecosystem functioning through a cascading effect on the microbial loop and the microbial food web.

### 4.1 Liming impact on DOM dynamics

Our data show the potential effects of hydrated lime addition on DOM dynamics determining a decrease in DOC concentration (Fig. 1 and 4) and a change in the optical properties of CDOM (Fig. 2, 3, 5 and 6). The decrease in $a_{254}$, the increase in $S_{275-295}$ (Fig. 2 and 5) and the decrease in humic-like fluorescence (Fig. 3 and 6) indicate a change in DOM

quality with a shift towards molecules with lower average molecular weight and aromaticity degree. These effects are
already visible at pH 9, but becomes relevant at pH 10. Different hypotheses can explain our results:
1) DOM reacts with $Ca(OH)_2$ and the largest and most aromatic molecules are oxidized to $CO_2$;
2) the largest and most aromatic molecules adsorb onto primary and secondary carbonate precipitates, that form
following the $Ca(OH)_2$ addition, and sink;
3) the largest and most aromatic molecules aggregate forming polymer gels or large colloidal material and sink.
Interestingly, a significant decrease in DOC concentration was observed only in the unmixed samples at the end of the
experiment (22 h after the addition, Fig. 1 and 4), DOC oxidation to $CO_2$ by reaction with $Ca(OH)_2$ (hypothesis 1) can
therefore be ruled out as a possible removal mechanism. The other 2 hypotheses remain plausible and are supported by
the available literature (Conzonno and Cirelli, 1995; Kaushal et al., 2020; Leenheer and Reddy, 2008; Pace et al., 2012).
In lake waters, DOM was observed to adsorb onto carbonate particles and co-precipitate with them; the use of $CaCO_3$
precipitation was indeed suggested as an efficient technique for DOM removal during drinking water treatment processes
(Leenheer and Reddy, 2008). The mechanism of DOM co-precipitation and/or physical incorporation into $CaCO_3$ is due
to the formation of insoluble calcium. This hypothesis is further supported by the observation of $CaCO_3$ precipitation
following the dissolution of hydrated lime, that was enhanced by the occurrence of nucleation surfaces as particles or
solid mineral phases in the solution (Moras et al., 2021).
In freshwater ponds, a high affinity of high molecular weight molecules to adsorb onto particles like $CaCO_3$ was observed
by Conzonno and Cirelli (1995) together with a preferential removal of high molecular weight humic substances during
$CaCO_3$ crystals formation. Since humic acids have important environmental functions in controlling the pH and the
bioavailability of dissolved metals (Baalousha et al., 2006), their removal may trigger a cascade effect with possible
impacts on water quality. Past studies showed that pH per se can affect DOM dynamics as DOM can undergo a fast
transition from dissolved to polymer gels (Chin et al., 1998) or large colloidal material (Pace et al., 2012) when pH
switches toward more basic values (pH > 8 for seawater).
Among the 3 hypotheses mentioned above, the decrease in $a_{254}$, observed in our experiments, supports the hypothesis 2,
suggesting that, following the addition of $Ca(OH)_2$, the largest and most aromatic dissolved organic molecules adsorb to
primary and secondary mineral particles and sink. This hypothesis is further supported by the high removal of the
terrestrial components observed in both the Mediterranean Sea ($C3_{Th\_Med}$, -26%) and the Baltic Sea ($C3_{Th-Bal}$, -13%). This
observation agrees with the results of Kaushal et al., (2020) which reported a higher incorporation of the terrestrial humic
substances into abiogenically precipitated aragonite, then transferred within coral skeletons, with respect to marine humic
substances.

**4.2 Different effects on Mediterranean and Baltic waters**
The Mediterranean Sea and the Baltic Sea are basins with different biogeochemical characteristics (Tab. 1). Our results
show that DOC concentration (Fig. 1 and 4) and $a_{254}$ (Fig. 2 and 5) are 6 and 13 times higher in the Baltic Sea than in the
Mediterranean Sea (Tab. 1), these data are consistent with previous studies (Hoikkala et al., 2015; Santinelli, 2015;
Santinelli et al., 2010). The lower $S_{275-295}$ (Tab. 1) and the different FDOM composition (Fig. S1 and S2) indicate a higher

percentage of terrestrial DOM in the Baltic Sea than in the Mediterranean Sea, as previously reported by Deutsch et al. (2012) and Hoikkala et al. (2015). Indeed, in the Baltic Sea, PARAFAC allowed to characterize humic-like and fulvic-like components but not protein-like ones (Fig. S2, Tab. S4), differently in the Mediterranean Sea the protein-like component was identified (Fig. S1, Tab. S3). Protein-like compounds are usually related to in-situ production, whereas fulvic-like substances mostly have a terrestrial origin. The predominance of terrestrial DOM in the Baltic Sea is due to the high freshwater input from the wide catchment area (~ 4 times as large as the sea itself), and the low seawater input from the North Sea. The Baltic Sea is also characterized by a peculiar carbonate system (Kuliński et al., 2017), exhibiting a wider range of total alkalinity (and pH) compared to the oceans. In particular, the Gulf of Riga, where the water for our experiment was collected, is characterized by a higher total alkalinity and a higher pH with respect to the rest of the Baltic Sea (Beldowski et al., 2010; Kuliński et al., 2017).

In our experiments, we observed a different impact of $Ca(OH)_2$ addition in the Mediterranean Sea and Baltic Sea. In the Mediterranean Sea, a DOC decrease of 6 and 13 µM was recorded at pH 9 and 10, respectively (Tab. S2), indicating a net removal up to 18% of the initial DOC (Fig. 1, Tab. S2). In the Baltic Sea, the maximum removal observed was 6% at pH 10, whereas no effect was recorded at pH 9 (Fig. 4).

Even if the salinity, being markedly lower in the Baltic Sea than in the Mediterranean Sea, is probably the main driver of the less pronounced effects on DOM dynamic, it cannot be excluded that the peculiar carbonate system combined with the different concentration and quality of DOM may have influenced the lower removal rates observed in our experiment. The influence of water chemistry is already evident by the 4-times lower amount of $Ca(OH)2$ needed to reach pH 9 and 10 in the Baltic Sea than in the Mediterranean Sea. Since $CaCO_3$ precipitation can be one of the main mechanisms explaining our results, the lower amount of $Ca(OH)_2$ added in the Baltic Sea can explain the lower decrease of DOC observed in this basin than in the Mediterranean Sea. At pH 10, the overall DOC removed in the Baltic Sea is larger (27 µM) than in the Mediterranean Sea (11 µM), despite the lower $Ca(OH)2$ added. This suggests a removal of 450 µmol of DOC per gram of $Ca(OH)2$ added in the Baltic Sea, and 44 µmol of DOC per gram of $Ca(OH)2$ added in the Mediterranean Sea. This observation can be explained by the predominance of terrestrial DOM in the Baltic Sea which was suggested to be preferentially removed during abiogenic precipitation of aragonite with respect to marine DOM (Kaushal et al., 2020).

It is noteworthy that DOM in the Mediterranean Sea and in the oceans shows a clear seasonal cycle, mostly attributed to the changes in temperature, water stratification and biological activity, affecting DOM concentration, optical properties and stoichiometry (Carlson and Hansell, 2015; Santinelli, 2015; Santinelli et al., 2013). Seasonality strongly affects DOM dynamics also in the Baltic Sea with prevalent allochthonous sources in winter and in-situ production by phytoplankton in spring (Hoikkala et al., 2012; Seidel et al., 2017). Our results, combined with the observed seasonality in DOM dynamics, stress that any plans for liming-based OAE should also take into consideration the season.

**4.3 Changed DOM dynamics: implication for the marine ecosystems**

Our results suggest that $CaCO_3$ precipitation is the main driver for the sequestration of DOM from the water column. The sinking of the largest and most complex fraction of DOM to the deep oceans could lead to different scenarios.

1. If the exported DOM is labile (i.e. it is available to microbial removal on the short temporal scale), its export would determine:

- A depletion of the energy available for heterotrophic prokaryotes in the surface layer, determining a malfunctioning of the microbial loop that could impact the energy transfer to the higher trophic levels. This process could be further enhanced if the primary production is limited by the reduced water transparency due to carbonate formation.
- The export of energy to the deepest layer (below the carbon compensation depth, CCD), leading to an increased bacterial production, in response to the labile DOM released due to the $CaCO_3$ dissolution.

2. If the exported DOM is refractory (i.e. it is not available to microbial removal on the short temporal scale), it will contribute to C sequestration in the deep waters.

Our results indicate the preferential removal of the humic-like fractions by $CaCO_3$ precipitation. Humic-like substances are considered to constitute the less labile fraction of DOM (Bachi et al., 2023; Zigah et al., 2017) , supporting C sequestration in the deep waters (hypothesis 2) and a change in the lability of DOM in the surface waters, with an increase in the percentage of the labile fraction after $CaCO_3$ formation. Even if the lability of DOM is a very complex process, depending on a large number of variables (Dittmar et al., 2021), the change in the lability of DOM could be tested in incubation experiments with natural microbial communities collected in the same area as the water used for the experiment. The water for the experiments was filtered through a 0.2 μm filter and it was therefore considered sterile, in order to investigate the potential removal of DOM by microbes, we could inoculate the natural microbial community adding a 10% of unfiltered water from the same site. In order to avoid artefacts from direct pH impacts on the microbial community, before the inoculum the pH should be brought to natural pH by adding HCl.

It should also be taken into consideration that the adsorption of DOM onto $CaCO_3$ particles, itself might reduce the bioavailability, regardless of the inherent properties of the DOM. This process would increase the carbon sequestration into the deep waters, but it would reduce the energy available for the marine ecosystems.

## 5. Conclusions

This study reports the first evidence of the potential effects of OAE on DOM dynamics in two contrasting environments: the oligotrophic Mediterranean Sea, known for its low DOC concentration, and the eutrophic Baltic Sea, characterized by high DOM concentration mostly of terrestrial origin. Our findings suggest that ocean alkalinization by $Ca(OH)_2$ sparging may alter DOM dynamics and, consequently, have a potential impact on the entire marine ecosystem. To mitigate these effects, it is crucial to reduce the duration and intensity of pH spikes, ensuring they remain below the safety threshold of pH 9. We stress the need to take into consideration the physico-chemical properties (e.g. salinity, pH, DOM concentration and quality) of the basin and the season, to efficiently manage ocean liming and mitigate the potential impacts of ocean alkalinization on DOM pool.

Although the experimental conditions used in this study were more severe than actual liming practices, where the release of $Ca(OH)_2$ in the ship's wake undergoes rapid dilution that significantly reduces pH changes, our results provide new insights into the possible impacts due to physico-chemical processes.

It is important to highlight that the experiments in this study were conducted using sterilized seawater, thus excluding the potential interplay of biological processes on DOM dynamics. To gain a more comprehensive understanding of possible OAE impacts, future research should address the influence of biological processes, as well as factors like dilution rates, water mixing, and realistic durations and severities of pH peaks. Scaling up the experimental setup to mesocosms would

allow for repeated additions and longer observation periods, enabling a more accurate representation of real-world
conditions.

**Author contribution**

Conceptualization by CS, DB and AA. CS designed and supervised the experiments and SV, RBS, GB, GC, MG carried them out.
Funding acquisition by SC and AA. CS prepared the manuscript with contributions from all co-authors.

**Competing interests**

The authors declare that they have no conflict of interest.

**Acknowledgements**

The current study has received funding from European Lime Association (EuLA) through a research contract established with CoNISMa
(National Inter-University Consortium for Marine Sciences) in Italy. We warmly thank Roberto Moreschi and Dario Ravasio
(UNICALCE Sedrina) for kindly sending the samples of Ca-hydroxide used for our experiments. We thank Giovanni Cappello
(Limenet) and Agija Bistere (Hyrogas) for sampling and sending to Pisa the Baltic Sea seawater. The authors are grateful to Marco
Carloni and Valtere Evangelista for their support in sampling of Mediterranean Sea water and CDOM/FDOM analyses and to Rosanna
Cascone, Rosanna Claps and Claudia Neri, (IBF-CNR, Italy) for the assistance in the financial management. The authors wish to thank
Aurela Shitza and Marlena Wissel from EuLA for their valuable feedback and helpful suggestions which greatly contributed to the
overall improvement of this paper.

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
