# Peer review of "Ocean liming effects on dissolved organic matter dynamics"

_EGUsphere, 2024_

## Author Comment (AC1)

**RC1: 'Comment on egusphere-2024-625', Anonymous Referee #1, reply**

The authors add hydrated lime to filter-sterilized seawater and track the changes in DOC concentration and optical parameters over a 24 hour period. The resulting conclusions provide a preliminary view into how dissolved organic carbon would change immediately following the addition of calcium hydroxide to seawater, work that is relevant for attempts to sequester carbon from the atmosphere. The conclusions from this work are limited, but could be a valid first step towards understanding this portion of the impact of liming on DOC.

We are grateful to the referee for her/his appreciation of our work. We are aware that the conclusions from this work are limited, but we strongly believe that this paper can bring new insights into the impact of ocean liming on DOM dynamics, and we believe that in this moment it is important to stress the need of further studies on this aspect. In the attached file you can find a point by point reply to all the issues raised by the referee.

The abstract discusses 'more pronounced effects', but the manuscript appears to have no statistical tests applied to support the conclusions presented by the authors.

All the differences, discussed in the paper, are supported by Kruskal–Wallis nonparametric test, as reported in the material and methods: "*For all parameters, differences were tested using the Kruskal–Wallis nonparametric test and were considered significant at the threshold of $p < 0.05$. All statistical analyses were performed using OriginPro version 9 (OriginLab, USA).*" In the Baltic Sea, no significant change was observed 3 hours after $Ca(OH)_2$ addition in both treatments (pH 9 and 10) (Fig. 4). At the end of the experiment (22 h), DOC decreased by 27 µM (7 %) at pH 10, whereas no significant change was observed at pH 9. In contrast, In the MedSea, despite the markedly lower DOC concentration (67 ± 2 µM). Three hours after $Ca(OH)_2$ addition, a 4 µM (6%) DOC decrease was observed in both treatments (Fig. 1). A further decrease was observed in the supernatant of the unmixed sample 22 h after the addition, with DOC reaching 59 ± 0.2 µM (12% decrease) at pH 9 and 56 ±1 µM (16% decrease) at pH 10. For these reasons, we wrote in the abstract: "*These effects, detectable at pH 9, become significant at pH 10 and are more pronounced in the Mediterranean than in the Baltic Sea.*"

I am confused a little about how pH was considered as a variable. I recognize that two different conditions were set up (pH = 9 and pH = 10). However, the CDOM/FDOM samples were then brought to neutral pH before analysis. Given that one key parameter under consideration is pH, I don't understand the justification to remove the impact of pH on the CDOM/FDOM measurement.

We thank the referee for this comment, since it highlights a problem, we had to deal with. Absorption and fluorescent measurements are affected by pH, so it makes no sense to compare spectra carried out in different pH conditions, for this reason, after an internal discussion, we decided to work at neutral pH. Changing the pH will not allow us to identify changes in CDOM and FDOM that are strictly dependent on pH, for optical properties of CDOM we will therefore only be able to study the impact of lime addition, not the impact of the consequent pH increase. However, as reported in the introduction: "*The aim of this study is to provide the first experimental evidence about the potential effects of hydrated lime addition on DOM dynamics in the oceans, by assessing changes in its concentration and optical properties (absorption and fluorescence)*." In the revised paper, we will modify the text accordingly.

The authors indicate that pH was measured, but the only pH data are presented for the initial conditions. Did the change in pH over the course of the incubation correspond to any of the changes in DOC/CDOM/FDOM? Given the importance of pH on the solubility of carbon in water this is an important parameter to consider over the time of the incubation.

The pH was indeed measured each time before subsampling as mentioned in the text. We apologize for not having clearly stated that it did not change during the 22h of the experiment. In the revised paper, we will change the text in the methods to clearly state that pH was stable $Ca(OH)_2$ addition and, in the supplementary material, we can add a summarizing table including pH data.

Abbreviating Mediterranean Sea and Baltic Sea seems unnecessary.

Ok, in the revised text we will avoid them.

Additional comments:

Line 33: 'These effects…'

Ok, in the revised text we will correct it.

Line 52: 'sharp increase in pH of about 1 unit, becoming lower than 0.2 units, 1400 – 1600 m far from the discharge site…' this is somewhat awkward and not clear. Please reword this sentence.

The sentence can be reworked as follows: "*Caserini et al. (2021) simulated the pH dynamics within the wake of a sparging ship releasing Ca(OH)2 with an initial particle radius of*

*45 µm at a rate of 10 kg s⁻¹. The results of their modeling study suggest that in these conditions a temporary, sharp increase in pH of about 1 unit can be observed at the discharge site, and that the effect decreases moving far from the discharge site, becoming lower than 0.2 pH units at a distance of 1400 – 1600 m (0.8-0.9 miles).*"

Line 102: I would alter the order information is presented in this paragraph. The first sentence indicates the bottles were mixed, and then the exception is given. This would be clearer if you start by indicating first that supernatant was collected from each bottle, then the bottles were mixed, and then an additional sample was collected.

In agreement with this comment, the sentence could be changed as follows: *"The bottles were gently mixed before subsampling at 5', 30', 3 h. After 22 hours, carbonate sedimentation was clearly visible at the bottom of the bottles, samples of the supernatant were therefore collected before mixing for both DOC and CDOM/FDOM analyses, an additional sample was collected after gently mixing only for DOC analyses since CDOM/FDOM would be strongly affected by the suspended particles."*

Line 106: the methods section here indicates the particles were removed by filtration, but line 166 indicates no mixed samples were collected. Please clarify.

We apologize for the misunderstanding. Since the particles strongly affect spectroscopic analyses, the mixed samples after 22h were collected only for DOC concentration measurements. Indeed, the results for 22h mixed samples are reported for DOC only (see figure 1 and 4). As reported at the previous point, the text can be changed as follows: "*The bottles were gently mixed before subsampling at 5', 30', 3 h. Since, after 22 hours, carbonate sedimentation was clearly visible at the bottom of the bottles, samples of the supernatant were collected before mixing for both DOC and CDOM/FDOM analyses, and an additional sample was collected after gently mixing only for DOC analyses since CDOM/FDOM would be strongly affected by the suspended particles.*"

Line 147: correct to Kruskal Wallis (only one L in Kruskal)

Ok, in the revised text we will correct it.

Line 178: correct to tryptophan

Ok, in the revised text we will correct it.

Line 272: Considering the rather low salinity of the Baltic Sea sample, the statements about the increased terrestrial DOM can be attributed to the fact that the sample was not a truly marine sample. Given the large salinity differences between the two samples, the conclusions about Mediterranean vs. Baltic might be a stretch as they could also be solely due to differences in salinity, as the authors elude to in this paragraph.

We agree with the referee that salinity can be one of the major factors affecting our results. However, given the complexity of the systems it seems difficult that salinity alone can explain the observed results. In the discussion, we highlighted the predominant effect of the salinity as follows: "*Even if the salinity, being markedly lower in the BalSea than in the MedSea, is probably the main driver of the lower precipitation of CaCO3, and consequently of the less pronounced effects on DOM dynamic, it cannot be excluded that […].*" We therefore think that we gave enough emphasis to the predominant role of salinity and we did not elude this aspect in this paragraph.

Line 293: Extrapolating conclusions to the importance of seasons is a bit of a stretch here, two samples collected one month apart do not allow any sort inference about the relevance of season. There are many additional parameters that would be needed before details on OAE would be clear, seasonality is but one.

We thank the referee for this comment, we totally agree with. Indeed, in the conclusions we highlighted all the parameters we need to know before details on OAE can be clear. In agreement with this comment, in the revised text we can integrate the last sentence of this paragraph in the conclusions, pointing out that seasonality should be one factor to take into consideration, in addition to all the others.

Figure 4: are there replicates for all the samples? If yes, please note that the error bars are smaller than the symbol.

Yes, there are 3 replicated for each sample. In the revised version, we can add the sentence: " *Please, note that for some samples the error bars are smaller than the symbols and therefore not visible.*", in the caption of figure 4.

Supplemental Table S1 – What is an 'EU representative product'? Is the product specification the desired set of parameters or the actual set of parameters? What is UniCalce (2021)? It is not in the list of references provided.

We thank the referee for raising this point and we really apologize for the unclear definition. "*EU representative product*" means that the properties of the $Ca(OH)_2$ used for the experiments are within the ranges that the EU defined for the $Ca(OH)_2$ production. For clarity, in the revised paper, we can replace 'EU representative product' by "$Ca(OH)_2$ properties" and modify the caption as follows: "*Properties of the high-purity calcium hydroxide used for the experiment. The product was provided by Unicalce ([https://www.unicalce.it/)](https://www.unicalce.it/)).*"

Table S2 – tryptophan (no e at the end)

Ok, in the revised text we will correct it.

---

## Author Comment (AC2)

**CC1:** 'Comment on egusphere-2024-625', **Meilian Chen,** reply

This manuscript presented ocean liming effects on DOC and optical properties of DOM. The lab controlled experiments are limited to two sets of samples from the Mediterranean Sea and the Baltic Sea following 1 day of sample liming. The treatments included adjustment of pH to 9 and 10. The topic is interesting since it's a potential solution for ocean acidification. The authors have seen changes o bulk DOC and also optical properties after liming. However, the experiment design excluded the interplay of other factors, such as sunlight, microbes, salinity, longer time. Those could be future studies.

We thank Dr. Meilian Chen for her appreciation of our work. We totally agree that this is only a first step in the study of the impact of ocean liming and that future studies are needed. However, we believe that this first evidence is important and may stimulate the discussion in the scientific community highlighting the need for additional studies.

Specific comments:

Table 1. The salinity of the Baltic Sea was only 6 and the $a_{254}$ was as high as 25 m$^{-1}$. It's more representative of an estuarine sample.

We agree with this comment, but these are the typical characteristics of the Baltic Sea. It was one of our goals to have very different environments in order to highlight the strong dependance of ocean liming effects on different waters (regions). Our results point out the need of broadening the study areas for future experiments to have a more global evaluation on the possible effects.

Lines # 104. The pH of the treated samples were brought to 7.5. Could this interfere with the liming treatment of DOM?

This is a very good point; pH affects spectroscopic analyses, so in order to have comparable data we needed to bring the pH to 7.5.

Lines #123 How about absorption coefficient at 280 nm or 320 nm?

We did not report all the absorption coefficients because they had the same trend and we wanted to keep the results as clear as possible without an overloading of data that do not add information. However, a table with all the data, including $a_{280}$ and $a_{320}$ can be added in the supplementary material of the revised paper.

Lines #126 Slope Ratio of S275-295 to S350-400 would be a better proxy of molecular weight.

We thank for this comment, the Sr can be included in the table that can be added in the supplementary material of the revised paper. We preferred to focus on $S_{275-295}$, since in the Med Sea absorption is very low at wavelengths higher than 350 and this could affect the Sr.

---

## Author Response (AR1)

**RC1: '[Comment on egusphere-2024-625'](), Anonymous Referee #1, [reply]()**

The authors add hydrated lime to filter-sterilized seawater and track the changes in DOC concentration and optical parameters over a 24 hour period. The resulting conclusions provide a preliminary view into how dissolved organic carbon would change immediately following the addition of calcium hydroxide to seawater, work that is relevant for attempts to sequester carbon from the atmosphere. The conclusions from this work are limited, but could be a valid first step towards understanding this portion of the impact of liming on DOC.

We are grateful to the referee for her/his appreciation of our work. We are aware that the conclusions from this work are limited, but we strongly believe that this paper can bring new insights into the impact of ocean liming on DOM dynamics, and we believe that in this moment it is important to stress the need of further studies on this aspect. Below, you can find a point by point reply to all the issues raised by the referee.

The abstract discusses 'more pronounced effects', but the manuscript appears to have no statistical tests applied to support the conclusions presented by the authors.

All the differences, discussed in the paper, are supported by Kruskal–Wallis nonparametric test, as reported in the material and methods: "*For all parameters, differences were tested using the Kruskal–Wallis nonparametric test and were considered significant at the threshold of $p < 0.05$. All statistical analyses were performed using OriginPro version 9 (OriginLab, USA).*" In the Baltic Sea, no significant change was observed 3 hours after $Ca(OH)_2$ addition in both treatments (pH 9 and 10) (Fig. 4, Tab. S2). At the end of the experiment (22 h), DOC decreased by 23 µM (6 %) at pH 10, whereas no significant change was observed at pH 9. In contrast, In the MedSea, despite the markedly lower DOC concentration (67 ± 2 µM). Three hours after $Ca(OH)_2$ addition, a 4 µM (6%) DOC decrease was observed in both treatments (Fig. 1). A significant decrease was observed in the supernatant of the unmixed sample 22 h after the addition, with DOC reaching 59.5 ± 0.1 µM (9% decrease) at pH 9 and 56.3 ±1.7 µM (18% decrease) at pH 10 (Tab. S2). For these reasons, we wrote in the abstract: "*These effects, detectable at pH 9, become significant at pH 10 and are more pronounced in the Mediterranean than in the Baltic Sea.*"

I am confused a little about how pH was considered as a variable. I recognize that two different conditions were set up (pH = 9 and pH = 10). However, the CDOM/FDOM samples were then brought to neutral pH before analysis. Given that one key parameter under consideration is

pH, I don't understand the justification to remove the impact of pH on the CDOM/FDOM measurement.

*We thank the referee for this comment, since it highlights a problem we had to deal with. Absorption and fluorescent measurements are affected by pH, so it makes no sense to compare spectra carried out in different pH conditions; for this reason, after an internal discussion, we decided to work at neutral pH. Changing the pH will not allow us to identify changes in CDOM and FDOM that are strictly dependent on pH, for optical properties of CDOM we therefore were only able to study the impact of lime addition, not the impact of the consequent pH increase. However, as reported in the introduction: "The aim of this study is to provide the first experimental evidence about the potential effects of hydrated lime addition on DOM dynamics in the oceans, by assessing changes in its concentration and optical properties (absorption and fluorescence)." In the revised paper, we modified the text accordingly.*

The authors indicate that pH was measured, but the only pH data are presented for the initial conditions. Did the change in pH over the course of the incubation correspond to any of the changes in DOC/CDOM/FDOM? Given the importance of pH on the solubility of carbon in water this is an important parameter to consider over the time of the incubation.

*The pH was indeed measured each time before subsampling as mentioned in the text. We apologize for not having clearly stated that it slightly changed during the experiment. In the revised paper, we added the following sentence in the material and methods section: "In the treatment at pH 9, the pH slightly decreased by 0.06 (Baltic Sea) and 0.29 (Mediterranean Sea) between 3 and 22 h after the addition (Tab. S2). In the treatment at pH 10, 3 hours after the addition the pH decreased by 0.3 in the Mediterranean Sea and after 22 hours it decreased by 0.45 in the Mediterranean Sea and 0.26 in the Baltic Sea (Tab. S2)." The pH data at each time are now reported in the supplementary material (Table S2).*

Abbreviating Mediterranean Sea and Baltic Sea seems unnecessary.

*Ok, in the revised text all the abbreviations were removed.*

Additional comments:

Line 33: 'These effects…'

*Ok, corrected.*

Line 52: 'sharp increase in pH of about 1 unit, becoming lower than 0.2 units, 1400 – 1600 m far from the discharge site…' this is somewhat awkward and not clear. Please reword this sentence.

We apologize for not being clear, in the revised text, the sentence was reworked as follows: *"Caserini et al. (2021) simulated the pH dynamics within the wake of a sparging ship releasing Ca(OH)$_2$ with an initial particle radius of 45 µm at a rate of 10 kg s$^{-1}$. The results of their modeling study suggest that in these conditions a temporary, sharp increase in pH of about 1 unit can be observed at the discharge site, and that the effects decrease moving far from the discharge site, becoming lower than 0.2 pH units at a distance of 1400 – 1600 m (0.8-0.9 nautical miles)."*

Line 102: I would alter the order information is presented in this paragraph. The first sentence indicates the bottles were mixed, and then the exception is given. This would be clearer if you start by indicating first that supernatant was collected from each bottle, then the bottles were mixed, and then an additional sample was collected.

In agreement with this comment, the sentence was changed as follows: *"The bottles were gently mixed before subsampling at 5', 30', 3 h. After 22 hours, carbonate sedimentation was clearly visible at the bottom of the bottles, samples of the supernatant were therefore collected before mixing for both DOC and CDOM/FDOM analyses, an additional sample was collected after gently mixing only for DOC analyses since CDOM/FDOM would have been strongly affected by the suspended particles."*

Line 106: the methods section here indicates the particles were removed by filtration, but line 166 indicates no mixed samples were collected. Please clarify.

We apologize for the not being clear. Since the scattering due to suspended particles strongly affects spectroscopic analyses , the mixed samples after 22h were collected only for DOC concentration measurements. Indeed, the results for 22h mixed samples are reported only for DOC (see figure 1 and 4). As reported at the previous point, the text was changed as follows: *"The bottles were gently mixed before subsampling at 5', 30', 3 h. After 22 hours, carbonate sedimentation was clearly visible at the bottom of the bottles, samples of the supernatant were therefore collected before mixing for both DOC and CDOM/FDOM analyses, an additional sample was collected after gently mixing only for DOC analyses since CDOM/FDOM would have been strongly affected by the suspended particles."*

Line 147: correct to Kruskal Wallis (only one L in Kruskal)

Ok, corrected.

Line 178: correct to tryptophan

Ok, corrected.

Line 272: Considering the rather low salinity of the Baltic Sea sample, the statements about the increased terrestrial DOM can be attributed to the fact that the sample was not a truly marine sample. Given the large salinity differences between the two samples, the conclusions about Mediterranean vs. Baltic might be a stretch as they could also be solely due to differences in salinity, as the authors elude to in this paragraph.

We agree with the referee that salinity can be one of the major factors affecting our results. However, given the complexity of the systems it seems difficult that salinity alone can explain the observed results. In the discussion, we highlighted the predominant effect of the salinity as follows: "*Even if the salinity, being markedly lower in the Baltic Sea than in the Mediterranean Sea, is probably the main driver of the lower precipitation of CaCO3 , and consequently of the less pronounced effects on DOM dynamic, it cannot be excluded that the peculiar carbonate system combined with the different concentration and quality of DOM may have influenced the lower removal rates observed in our experiment. The influence of water chemistry is already evident by the 4-times lower amount of Ca(OH)2 needed to reach pH 9 and 10 in the Baltic Sea than in the Mediterranean Sea. Since CaCO3 precipitation can be one of the main mechanisms explaining our results, the lower amount of Ca(OH)2 added in the Baltic Sea can explain the lower decrease of DOC observed in this basin than in the Mediterranean Sea.*" We therefore think that we gave enough emphasis to the predominant role of salinity and we did not elude this aspect in this paragraph.

Line 293: Extrapolating conclusions to the importance of seasons is a bit of a stretch here, two samples collected one month apart do not allow any sort inference about the relevance of season. There are many additional parameters that would be needed before details on OAE would be clear, seasonality is but one.

We thank the referee for this comment, we totally agree with. Indeed we only hypothesize the relevance of the season, that is reported to affect DOM concentration and quality, but of course our data cannot support any impact of season. In the conclusions we highlighted all the parameters we need to know before details on OAE can be clear. In agreement with this comment, in the revised text the last sentence of the paragraph was modified as follows: "*Our results combined with the observed seasonality in DOM dynamics stress that any plans for liming-based OAE should also take into consideration the season.*"

Figure 4: are there replicates for all the samples? If yes, please note that the error bars are smaller than the symbol.

Yes, there are 3 replicated for each sample. In the revised version, the following sentence was added in the captions of the figures: "*Please, note that for some samples the error bars are smaller than the symbols and therefore not visible*."

Supplemental Table S1 – What is an 'EU representative product'? Is the product specification the desired set of parameters or the actual set of parameters? What is UniCalce (2021)? It is not in the list of references provided.

We thank the referee for raising this point and we really apologize for the unclear definition. "*EU representative product*" means that the properties of the $Ca(OH)_2$ used for the experiments are within the ranges that the EU defined for the $Ca(OH)_2$ production. For clarity, in the revised paper, we replaced 'EU representative product' by "Hydrated lime properties" and the caption was modified as follows: "*Properties of the high-purity calcium hydroxide used for the experiment. The product was provided by Unicalce (https://www.unicalce.it/)*."

Table S2 – tryptophan (no e at the end)

Ok, corrected

The manuscript by Santinelli et al. tests how ocean alkalinity enhancement (OAE) affects DOM composition and concentration. They conducted an experiment with two seawater samples, in which Ca(OH)2 was added to pH of 9 or 10, and they found that in both samples, there was removal of DOC and of CDOM / FDOM at the highest pH treatment. Possible reasons are discussed, and adsorption to secondary carbonate minerals is identified as the most likely explanation.

Overall, the scope of the manuscript is a bit limited in only having experimented with two different water samples. However, I think that the novelty and timeliness of the study are both very high, and the two water samples cover two extremes in terms of DOM source and composition, so I think that the results are likely to be generally representative of effects from lime addition. I think that the experiments are fundamentally robust, even though clearly more research is needed to understand the underlying mechanisms and work out the implications. But I support publishing this manuscript already, because that will motivate further work in this direction. I have a number of relatively minor suggestions for revisions as listed below.

We are gratefully at the referee for her/his appreciation of our work and for having completely understood our point. We are aware of the limits of our experiments, but we really hope that this can be a stimulus for investigating in depth the effects of ocean liming on marine ecosystems, focusing also on chemical processes that affect the microbial world. We also thank the referee for the very constructive comments and for the suggestions.

Line 23: you say that DOM is "the largest … mixture of organic molecules on Earth". I suppose it depends a bit on how one wants to define individual pools, but collectively the soil organic carbon pool on land is at least twice as large as the DOC pool, so perhaps "one of the largest" would be better?

Yes, the sentence was modified as suggested.

Lines 52 and 74-75: Initially you talk about enhancement by +1 pH unit and release of 10 kg/s, but then the experimental treatments are +1 and +2 pH units based on release of up to 25 kg/s, but the same reference is given for both. Maybe explain this a bit more clearly in the introduction so it doesn't come across as being inconsistent?

We thank the referee for this comment, and we apologize for not being clear. The paper by Caserini et al. (2021) does not report a release of 25 kg/s, the number (10 kg/s) reported in the introduction is correct, but we realized that the sentence was not clear. In the revised text,

the sentence was reworked as follows: "*Caserini et al., (2021) simulated the pH dynamics within the wake of a sparging ship releasing Ca(OH)2 with an initial particle radius of 45 μm at a rate of 10 kg s-1. The results of their modeling study suggest that in these conditions a temporary, sharp increase in pH of about 1 unit can be observed at the discharge site, and that the effects decreases moving far from the discharge site, becoming lower than 0.2 pH units at a distance of 1400 – 1600 m (0.8-0.9 miles).*"

We also reworked the sentence in the methods as follows: "*Based on the results by Caserini et al., (2021), which suggested a sharp increase of 1 unit of pH at the discharge site of a sparging ship, the experiment was carried out at pH 9. Although unlikely under actual conditions of dilution in the open sea, an additional experiment was carried out at pH 10 because this situation may occur in coastal waters (e.g. coastal lagoons, high primary productivity enhanced by eutrophication; Hinga, 2002)*".

**Reference**: K.R. Hinga, 2002. Effects of pH on coastal marine phytoplankton. Mar Ecol Prog Ser 238: 281 - 300, 2002

Line 132: "The EEMs were elaborated" – better to say "processed" or "analysed".

Ok, elaborated was replaced by processed.

Line 138-139: I don't understand well how the sample sizes come about, and especially why the sample size would be different, given that the experimental design is the same. I suggest explaining this here. Based on Section 2.1, I would expect a sample size of (2 pH treatments + 1 control) x (4 time points) x (3 replicates) = 36 samples for the CDOM/FDOM analysis.

We thank the referee for notice this mistake and we apologize for the misprint. The correct number of samples is 45 for each experiment. Time points are 5 because we also considered the t0 taken before the Ca(OH)$_2$ addition. The correct number is therefore (2 pH treatments + 1 control) x (5 time points) x (3 replicates) = 45 samples. We changed the text accordingly. Some EEMs were measured more than 1 time, this explains the different numbers in the previous version.

Lines 247–263: This is an interesting discussion. Another study that might be relevant here is Kaushal et al. (2020), who conducted aragonite precipitation experiments with seawater and different DOM sources to investigate humic substance incorporation into coral skeletons (for transparency: I am a co-author on that study). In that case, the evidence suggested preferential incorporation of terrestrial humic substances, which I think is consistent with the results here,

and more generally helps explain incorporation of FDOM into coral skeletons. The literature cited here at the moment seems to be mostly on freshwater lakes/ponds.

We really thank the referee for this comment and for suggesting this interesting paper. In the revised text, this point was included as follows: "*Among the 3 hypotheses mentioned above, the decrease in a254, observed in our experiments, supports the hypothesis 2, suggesting that, following the addition of Ca(OH)2, the largest and most aromatic dissolved organic molecules adsorb to primary and secondary mineral particles and sink. This hypothesis is also supported by the higher removal of the terrestrial components in both the Mediterranean Sea (C3Th_Med, -26%) and the Baltic Sea (C3Th-Bal, -11%). This observation is in agreement with the results of (Kaushal et al., 2020) which observed a higher incorporation of terrestrial humic substances into abiogenically precipitated aragonite, with respect to marine-humic ones.*"

Lines 186-292: It's clear that in the Baltic sample, a smaller percentage of the initial DOC pool was removed, but I think it's important to also recognise that this represents a much larger absolute quantity of DOC removed. I didn't see data presented anywhere on the amount of CaCO3 being formed, but the first sentence here suggests that there was less CaCO3 formed in the Baltic experiment (please clarify whether you have data to show that less carbonate was formed, or whether this is an assumption). If the amount of CaCO3 that was produced in the Baltic samples was equal to or less than the amount in the Mediterranean, then that means that per mass of CaCO3 the Baltic DOM is being strongly preferentially removed compared to the DOM in the Mediterranean. This is consistent with the conclusion in Kaushal et al. (2020) that CaCO3 preferentially removes terrestrial FDOM.

Unfortunately, we did not measure the amount of $CaCO_3$ formed in the experiments, it was an assumption due to the lower amount of Ca(OH)$_2$ added. We therefore modified the sentence as follows: "Even if the salinity, being markedly lower in the Baltic Sea than in the Mediterranean Sea, is probably the main driver of  the less pronounced effects on DOM dynamic, […]".

However, even if the amount of $CaCO_3$ formed is unknown, in the revised paper, we added the following paragraph taking into consideration the overall amount of DOC removed in the two experiment, together with the amount of Ca(OH)$_2$ added: "*At pH 10, the overall DOC removed in the Baltic Sea is larger (27 µM) than in the Mediterranean Sea (11 µM), despite the lower Ca(OH)$_2$ added. This suggests a removal of 450 µmol of DOC per gram of Ca(OH)$_2$ added in the Baltic Sea, and 44 µmol of DOC per gram of Ca(OH)$_2$ added in the Mediterranean Sea. This observation can be explained by the predominance of terrestrial DOM in the Baltic Sea which was suggested to be preferentially removed during abiogenic precipitation of aragonite with respect to marine DOM (Kaushal et al., 2020).*"

Line 298: "any hypothesis of liming-based OAE" is unclear. Do you mean "any proposal" or "any plans for"?

Yes, in the revised text "*hypothesis*" was replaced by "*any plans for*"

Section 4.3: This section might be improved with a bit more detailed discussion. I appreciate that this will necessarily be a bit speculative, but the main change seems to be in the humic and fulvic fractions, which typically not the most highly labile parts of the DOM pool. At the same time, if the DOM is sorbed onto CaCO3 particles, that in itself might alter the bioavailability, regardless of the inherent properties of the DOM. You could therefore discuss in a few sentences which of these scenarios you think is the more probable one. Perhaps even more important would be if you could provide some thoughts on the design of experiments to test this, as it would be important to avoid artefacts from direct pH impacts on the microbial community.

Kaushal et al. (2020). Sub-annual fluorescence measurements of coral skeleton: relationship between skeletal luminescence and terrestrial humic-like substances. Coral Reefs 39:1257–1272. https://doi.org/10.1007/s00338-020-01959-x

We thank for this comment and for pushing us to improve this section, even if in a speculative way. In the discussion of the revised paper, we added the following paragraph: "*Our results indicate the preferential removal of the humic-like fractions by CaCO3 precipitation. Humic-like substances are considered to constitute the less labile fraction of DOM (Zigah et al. 2017; Bachi et al., 2023), supporting C sequestration in the deep waters (hypothesis 2) and a change in the lability of DOM in the surface waters, with an increase in the percentage of the labile fraction after CaCO3 formation. Even if the lability of DOM is a very complex process, depending on a large number of variables (Dittmar et al., 2021), the change in the lability of DOM could be tested in incubation experiments with natural microbial communities collected in the same area as the water used for the experiment. Since the water for the experiment is filtered through a 0.2 µm filter and it is considered sterile, so we could add unfiltered water from the same site to add the natural microbial community. In order to avoid artefacts from direct pH impacts on the microbial community, before the inoculum the pH should be brought to natural pH by adding HCl.*

*It should also be taken into consideration that the adsorption of DOM onto CaCO3 particles, itself might reduce the bioavailability, regardless of the inherent properties of the DOM. This process would increase the carbon sequestration into the deep waters, but it would reduce*

*the energy available for the marine ecosystems, since it would increase DOM persistence in deep waters."*

**CC1: 'Comment on egusphere-2024-625', Meilian Chen, reply**

This manuscript presented ocean liming effects on DOC and optical properties of DOM. The lab controlled experiments are limited to two sets of samples from the Mediterranean Sea and the Baltic Sea following 1 day of sample liming. The treatments included adjustment of pH to 9 and 10. The topic is interesting since it's a potential solution for ocean acidification. The authors have seen changes o bulk DOC and also optical properties after liming. However, the experiment design excluded the interplay of other factors, such as sunlight, microbes, salinity, longer time. Those could be future studies.

We thank Dr. Meilian Chen for her appreciation of our work. We totally agree that this is only a first step in the study of the impact of ocean liming on DOM dynamics and that future studies are needed. However, we believe that this first evidence is important and may stimulate the discussion in the scientific community highlighting the need for additional studies.

Specific comments:

Table 1. The salinity of the Baltic Sea was only 6 and the $a_{254}$ was as high as 25 m$^{-1}$. It's more representative of an estuarine sample.

We agree with this comment, but these are the typical characteristics of the Baltic Sea. It was one of our goals to have very different environments in order to highlight the strong dependance of ocean liming effects on different waters (regions). Our results point out the need of broadening the study areas for future experiments to have a more global evaluation on the possible effects.

Lines # 104. The pH of the treated samples were brought to 7.5. Could this interfere with the liming treatment of DOM?

This is a very good point and we thank Dr. Meilian Chen to raise it. Absorption and fluorescent measurements are affected by pH, so it makes no sense to compare spectra carried out in different pH conditions; for this reason, after an internal discussion, we decided to work at neutral pH. Changing the pH will not allow us to identify changes in CDOM and FDOM that are strictly dependent on pH, for optical properties of CDOM we were therefore only able to study the impact of lime addition, not the impact of the consequent pH increase. However, as reported

in the introduction: "*The aim of this study is to provide the first experimental evidence about the potential effects of hydrated lime addition on DOM dynamics in the oceans, by assessing changes in its concentration and optical properties (absorption and fluorescence)*." In the revised paper, we modified the text accordingly.

Lines #123 How about absorption coefficient at 280 nm or 320 nm?

We did not report all the absorption coefficients because they had the same trend and we wanted to keep the results as clear as possible without an overloading of data that do not add information. In agreement with this comment, a table with all the data, including $a_{280}$ and $a_{320}$, is added in the supplementary material of the revised paper (Table S2).

Lines #126 Slope Ratio of S275-295 to S350-400 would be a better proxy of molecular weight.

We thank for this comment, the Sr is included in the table added in the supplementary material of the revised paper (Table S2). In the text, we preferred to focus on $S_{275-295}$, since in the Mediterranean Sea absorption is very low at wavelengths higher than 350 and this could affect the Sr.